

# Can adaptations of crop and soil management prevent yield losses during water scarcity? - A modelling study

Malve Heinz[1,2,3], Maria Eliza Turek[1,2], Bettina Schaefli[1,3], Andreas Keiser[4], and Annelie Holzkämper[1,2]

[1]Oeschger Centre for Climate Change Research, University of Bern, Switzerland

[2]Agroecology and Environment, Agroscope, Switzerland

[3]Hydrology, Institute of Geography, University of Bern, Switzerland

[4]Arable farming and plant breeding, School of Agricultural, Forest and Food Sciences HAFL, Bern University of Applied Sciences, Switzerland

**Correspondence:** Malve Heinz (malve.heinz@unibe.ch)

**Abstract.** With climate change, the increasingly limited availability of irrigation water resources poses a major threat to agricultural production systems world-wide. This study explores climate adaptation options in soil and crop management to reduce yield losses due to water scarcity and irrigation restrictions during the 2022 summer drought. The focus is on potato production in the Broye catchment in Switzerland, which is representative of many mid-sized lowland catchments in Central Europe facing

reduced irrigation water availability. We employed the field-scale agro-hydrological model SWAP in a distributed manner to simulate regional irrigation demand, yields and deficits under drought stress. Results suggest that irrigation bans and drought in 2022 led to a 16.4% reduction in potato yield due to a 59% deficit in irrigation water. Our findings suggest that adding 1% soil organic carbon (SOC) down to a depth of 60 cm could have reduced the yield loss to only 7%. Planting earlier maturing potato varieties in less favorable pedoclimatic conditions further improves irrigation water productivity (IWP) and reduces irrigation

water demand by 26%. In this case, however, there is a trade-off in yield, the reduction of which can only be reduced to -14.8%. Overall, our findings highlight the great value of soil organic carbon for preventing productivity losses during droughts at the example of a recently experienced drought year. Furthermore, we show that irrigation water use efficiency can be optimized by location-specific combinations of adaptation choices. In the face of future droughts exacerbated by climate change, the measures studied here represent a valuable adaptation to mitigate yield losses and reduce dependence on irrigation.

**Keywords.** climate change; drought; climate adaptation; soil carbon sequestration; SWAP; agro-hydrological modeling; Western Switzerland

## 1 Introduction

The agricultural sector is particularly vulnerable to climate change impacts, confronted with rising temperatures and shifting precipitation patterns that trigger agricultural and hydrological droughts (Fahad et al., 2017; IPCC, 2023; Uniyal and Dietrich,

2021). The projections for the future indicate a substantial increase in drought frequency, often compounded by heat waves, exacerbating the impacts of drought and heat stress on crops (IPCC, 2023). Elevated temperatures drive up potential evapotranspiration, amplifying crop water requirements (Allani et al., 2020). Simultaneously, ongoing and projected decreases of



summer precipitation in regions across the mid- and high-latitudes amplify the demand for irrigation to satisfy these increasing

crop water requirements (Allani et al., 2020). Following the drought in the summer of 2022, maize and soy yields in Europe

declined by 16% and 15% (Toreti et al., 2022a, b). Compared to current conditions, Leng and Hall (2019) project accelerated

global yield losses for major crops. For wheat and maize, an additional reduction of up to 12% and up to 6.3% is expected until

the end of the century, compared to 1961-2016, if no adaptation measures are taken (Leng and Hall, 2019). Given that droughts

are the leading cause of yield losses globally (Bodner et al., 2015), the imperative to enhance water utilization efficiency and

adapt water and soil management practices becomes increasingly evident (Bodner et al., 2015; Fahad et al., 2017).

Expanding irrigation to mitigate droughts may lead to distributional and prioritization conflicts among various water users

and aquatic ecosystems (He et al., 2023; Kreins et al., 2015). Such conflicts are reinforced in locations where climate change

affects seasonal water availability. In alpine regions, as warming progresses, more precipitation will fall as rain instead of

snow (Yang et al., 2021), resulting in reduced snowmelt-driven streamflow during the subsequent melt period (spring and early

summer). In pluvial lowland catchments, winter runoff will increase. At the same time, however, summer low flow is expected

to continue to decrease as increasing evapotranspiration rates meet decreasing seasonal precipitation, as in most of central

continental Europe (Floriancic et al., 2021). Consequently, irrigation will not always be feasible or able to alleviate the drought

and heat stress for crops. Therefore, it is crucial to implement strategies that i) reduce reliance on irrigation, ii) maintain soil

moisture for crops at sufficiently high levels for more extended periods, and iii) minimize losses through evaporation or surface

and subsurface runoff (Bodner et al., 2015).

Irrigation is often critical at specific stages of growth when the plant is especially prone to drought or heat stress induced

by inadequate soil moisture levels and lacking precipitation. Earlier maturing varieties can partially alleviate this issue by

accelerating phenological development to "escape" the drought. Planting early-maturing crops or winter varieties is often

recommended in temperate regions, such as Switzerland (Federal Office for the Environment (FOEN), 2019, 2021). It must,

of course, be noted that different varieties also lead to different properties and uses. This is, for example, the case for potatoes,

where late-maturing varieties are more suitable for processing (chips or fries), and early varieties lead to lower yields and

can only be used as table potatoes. In addition to the choice of crop variety, soil management practices, like cover cropping,

mulching, conservation tillage and organic amendments, can help boost soil moisture content (Diacono and Montemurro, 2010;

Hou et al., 2012; Kader et al., 2019; Moussa et al., 2002; Mulumba and Lal, 2008; Wezel et al., 2014).

    Organic amendments that target the increase of soil organic carbon (SOC) lead to an increased water retention and water-use

efficiency (Diacono and Montemurro, 2010; Eden et al., 2017). Turek et al. (2023) tested different levels and depths of increase

in soil organic carbon (SOC) and their impacts on crop transpiration. The study reports that adding 2% SOC until 60 cm depth

can reduce water stress and increase the resilience of crops at the onset of droughts. An increase in SOC can be achieved by

several practices such as cover cropping or the application of compost or manure. Cover crops grown over the winter between

the main summer crops can reduce surface runoff, protect the soil from erosion, increase SOC content and promote infiltration,

ideally making the water available for the main crops (Bodner et al., 2015; Wezel et al., 2014).

    These practices directly influence essential soil structural properties and associated soil physical parameters (namely hy-

draulic conductivity and water retention capacity), thereby modifying associated hydrological processes, such as evapotran-





spiration, infiltration, local surface and subsurface runoff formation, and groundwater recharge. In their extensive review on the impact of soil health measures on water resources in irrigation, Acevedo et al. (2022) give a broad overview of the prin-

ciples and practices of soil health and list future opportunities and challenges. They draw attention to the role of soil health in increasing green water consumption (share of crop evapotranspiration that is satisfied by soil water provided by precipitation) and reducing the reliance on blue water (irrigation water taken from surface or groundwater, needed to bridge potential deficits). Acevedo et al. (2022) argued that while there is a large body of work concerning individual or sometimes combined soil management measures, the representation and translation into agro-hydrological models to quantify their impact on green

and blue water beyond field scale is still missing.

There are guidelines and studies on how to irrigate certain crops (Wriedt et al., 2009; Gu et al., 2020; Fricke and Riedel, 2019) or how to increase the efficiency of irrigation (Lalehzari and Kerachian, 2020; Maier and Dietrich, 2016). However, quantifying irrigation demand and supply deficits on a scale relevant to stakeholders such as infrastructure planners is complex and, therefore, rarely undertaken. This is a significant knowledge gap given that irrigation remains the world's largest user

of fresh water (Acevedo et al., 2022; Kennan et al., 2019; Samimi et al., 2020). Regional water demand serves as a pivotal metric for decision-making regarding water allocation and the strategic planning of new irrigation infrastructure projects, such as water retention basins. In their comprehensive review, Uniyal and Dietrich (2021) examined various approaches and model choices for estimating catchment-scale irrigation demand.

Uniyal and Dietrich (2021) evaluated various agro-hydrological models, their characteristics and their ability to represent

both hydrological and agricultural processes. Included are distributed models of different complexity that operate on a catchment scale, like SWAT (Arnold et al., 2012), WaSiM (Schulla, 2021), HYPE (Swedish Meteorological and Hydrological Institute (SMHI), 2023) or WEAP (Stockholm Environment Institute (SEI), 2015), but also the 1-D (soil-column scale) models like SWAP (Kroes et al., 2000) or AquaCrop (Food and Agriculture Organization of the United Nation (FAO), 2016) that can be upscaled and regionalized. They further differentiated mechanistic models and conceptual models. While mechanistic mod-

els like SWAP or WaSiM include and represent physical process descriptions, e.g., Richard's equation, models like SWAT or HYPE operate in a more simplistic way (Uniyal and Dietrich, 2021). Most catchment-scale agro-hydrological models employ a rough and static representation of crops and use the same parameterizations for entire crop classes (Uniyal and Dietrich, 2021; Zhang et al., 2021). When it comes to irrigation management, most models can schedule applications. Still, few can represent the water sources used for irrigation (surface water, groundwater) or their availability through time. The field-scale

model SWAP is coupled with the comprehensive crop-growth model WOFOST and, therefore, can dynamically simulate crop growth in response to primarily biogeochemical processes in a detailed way (Uniyal and Dietrich, 2021). Depending on the water source, SWAP can also represent water-use constraints in a simplified way by reducing the time frame when irrigation is feasible or authorized. SWAP is widely used in water management and climate change impact studies. Cyano et al. (2007) applied SWAP to evaluate the impacts of climate change on crop growth (maize and wheat) and irrigation requirements in

Turkey. Using SWAP, Utset et al. (2007) modeled sugar beet water use in the Mediterranean region. They pointed out the importance of calibrating and validating the standard parameters of the model to adapt them to local conditions, especially concerning temperature sums, which strongly influence development stages. Noory et al. (2011) used the integrated SWAP-



WOFOST model to assess measures to improve water management in Iran. To capture its heterogeneity, they divided the region into homogeneous simulation units with specific climatic, edaphic, and irrigation data. Despite patchy land use data, the model

achieved good accuracy in simulating annual surface runoff, although with a slight underestimation. Since irrigation scheduling can be automatized depending on different thresholds in soil moisture, the model is well suited to represent realistic irrigation applications and demand.

Winter et al. (2017) explored how agricultural assessments represent water scarcity. They concluded that when quantifying irrigation demand, the actual supply is often not given, and the impact of water availability on crop yield is not assessed. They

advocated using loosely coupled crop and hydrological models to fully capture irrigated agricultural systems. The restriction of water use is often not considered in models, which Winter et al. (2017) identified as a limitation of the state-of-the-art models. (Brochet et al., 2024) used SWAT to integrate irrigation water withdrawals into their streamflow predictions to account for the significant anthropogenic influence in low-flow periods. In their study area, daily irrigation water withdrawal was measured and used for calibration and validation. However, they did not take into account water restrictions, which are described to be

common and influential in the catchment.

Gorguner and Kavvas (2020) calculated the water balance in a semi-arid catchment to quantify the future unmet irrigation demands under climate change and used the FAO56 approach (Allen et al., 1998) to estimate irrigation water requirements. They found that future water levels in the reservoirs are insufficient to meet the increase in irrigation water demand under future climate change (Gorguner and Kavvas, 2020). Such efforts to quantify current or future irrigation water demand and

shortage have either been made globally (Wada et al., 2014; Müller Schmied et al., 2021; Joseph et al., 2020), in semi-arid catchments (Gorguner and Kavvas, 2020) or without considering water restrictions (Brochet et al., 2024; Masia et al., 2021). To our knowledge, no study has simulated irrigation demand while considering restrictions on water supply and the impact of these on yield for temperate regions.

In this context, this study presents an approach to reduce critical knowledge gaps by i) quantitatively assessing the impacts

of water resource limitations on agricultural productivity and yield losses during a recent drought year and ii) evaluating the benefits of soil and crop management adaptations to reduce such yield losses in the face of limited irrigation water resources. The selected case study is the Broye catchment in Switzerland, which is representative of many mid-sized lowland catchments in Central Europe, which experience a reduction in low flows and, therefore, in the irrigation water supply. Many catchments, like the Broye, are thus subject to temporary irrigation bans, impacting agricultural production.

The applied model is the field-scale SWAP (Soil-Water-atmosphere-Plant) model (Kroes et al., 2000), which is used here in a spatially explicit manner (Section 3.6.2). This physically-based model is suitable for this study because it simulates crop growth, irrigation demand and soil water fluxes in detail and can integrate irrigation water constraints through the irrigation module implemented within the WOFOST crop module (World-Food-Studies, Supit and Van Diepen (1994)). We evaluate the effects of crop and soil management in the form of earlier maturing varieties and increased soil organic carbon (SOC) on potato

irrigation demand and yield.



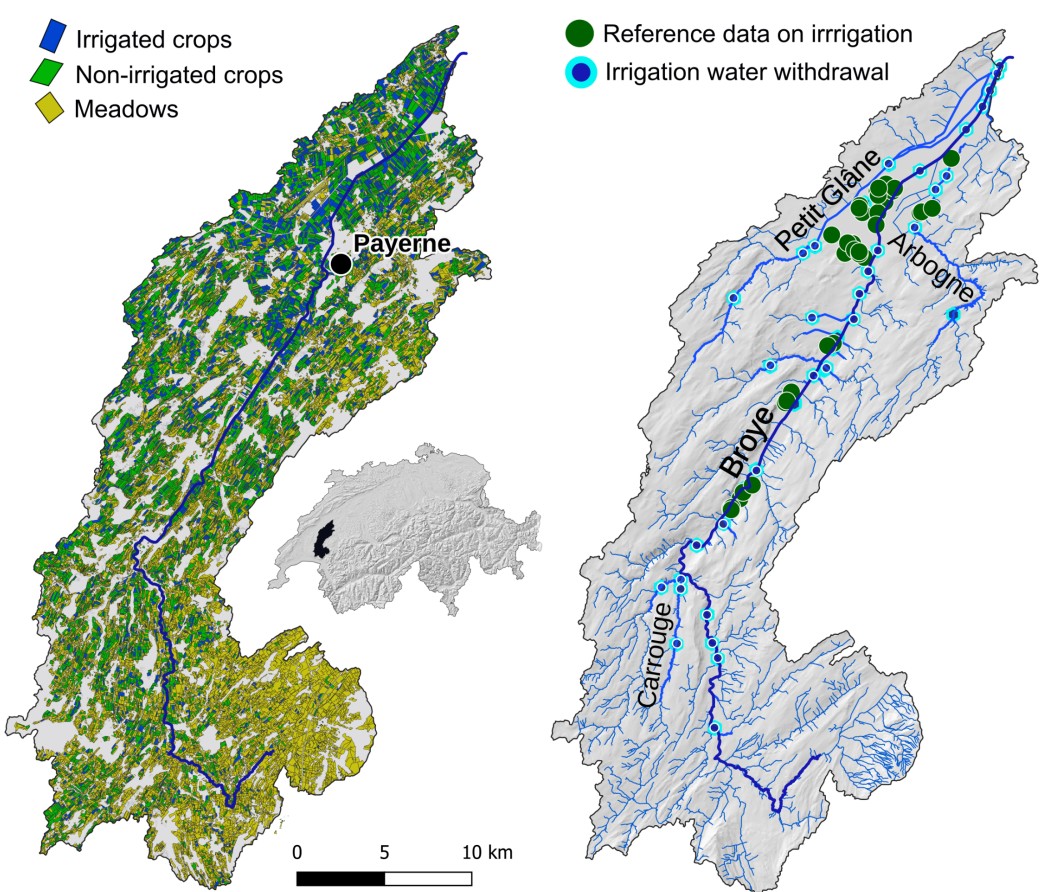

**Figure 1.** Left: Broye catchment and land use (KGK-CGC, 2023). Blue fields= irrigated crops, green fields = other arable crops, yellow fields= meadows. Right: Broye catchment with the Broye River and tributaries. Green dots = station-years reference data points for irrigation (School of Agriculture, Forest and Food Sciences HAFL, 2022). Blue points = concessioned withdrawal stations for irrigation water, locations provided by the respective cantons as considered in the report of HAFL (School of Agricultural, Forest and Food Sciences HAFL, 2023).

## 2 Study site

The Broye catchment is located in the South-Western part of the Swiss Central Plateau and covers an area of 604 km² with a maximum elevation of 1574 m asl (Hydrological Atlas of Switzerland (HADES), 2024). The Broye River has its source in the Fribourg Prealps and flows into Lake Murten at an altitude of 644 m asl (Hydrological Atlas of Switzerland (HADES), 2024).

The predominant soil types in the region are loam, clay loam and sandy loam (Figure 2, Swiss Competence Centre for Soil 2023), and the mean SOC content is 1.7%. The Broye catchment, being a part of the Swiss Central Plateau and a primary agricultural production zone, comprises 68% agricultural land in 2022 (Figure 1, KGK-CGC (2023).





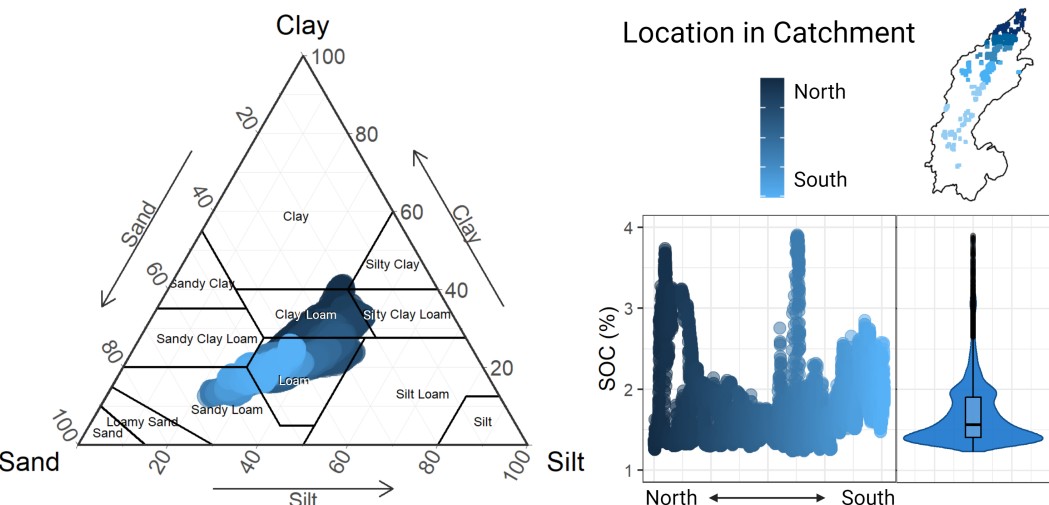

**Figure 2.** Soil texture and soil organic carbon (SOC) content over the study perimeter. Color gradient corresponds to the longitudinal gradient within the study perimeter (section 3.4.1)

The mean daily temperature in the catchment is 9°C, and the mean annual precipitation is 1158 mm (1991-2020; MeteoSwiss (2021b, a)). In 2022, the mean temperature was 10.9 °C, and the annual precipitation was 1003 mm. For the meteorological
station in Payerne, located within the main agricultural zone, the mean temperature was 11.4°C and annual precipitation 816 mm (MeteoSwiss, 2024).

The average daily discharge of the Broye is 7.7 $m^3s^{-1}$ (period 1920 to 2019), with a maximum monthly mean of 11 $m^3s^{-1}$ in March and a minimum monthly mean of 4.1 $m^3s^{-1}$ in August (Federal Office for the Environment (FOEN), 2023). Many lowland streams in western Switzerland, such as the Broye, experience extremely low flows during summer and autumn droughts.
In drought years, including 2003, 2015 and 2018, the annual discharge of the Broye and the monthly discharges from June to October were well below the long-term average (7.7 $m^3s^{-1}$, Federal Office for the Environment (FOEN) (2015, 2017, 2020)). Climate change is projected to reduce Swiss summer streamflow on average by up 20% (Federal Office for the Environment (FOEN), 2021). For lowland catchments, such as the Broye (mean elevation < 1500 m asl), a reduction by up to 50% by the end of the century is projected (Federal Office for the Environment (FOEN), 2021).
83% of the irrigation water is withdrawn with mobile pumps from the Broye and the Petit Glâne (a tributary to the Broye), the remaining 17% from other smaller tributaries and local water distribution networks (Robra and Mastrullo, 2011). Farmers have to apply for concessions to be allowed to irrigate in certain places, and sometimes these concessions are shared. In a survey from 2011, Robra and Mastrullo (2011) found that 89% of the area is irrigated with sprinkler irrigation, 6% with linear irrigation (long mobile pipes that distribute water across the field), and only 4.5% with drip irrigation (water is directly applied
to the roots).

The withdrawals for irrigation are prohibited when the Broye in Payerne drops below the legal minimum environmental flow, which in Switzerland is fixed to the 5th long term streamflow percentile. For the Broye, this value equals 1.26 $m^3s^{-1}$





(Robra and Mastrullo, 2011). From 2011-2022, water withdrawal bans from the Broye (and relevant tributaries) occurred in 9
years, beginning most often in mid-July and lasting until November (Châtelain, 2023).

The main crops grown in this catchment are winter wheat (33%), green and silage maize (13%), winter rapeseed (11%),
winter barley (9%), sugar beet (8%), and potatoes (5%) (Federal Office for Agriculture (FOAG), 2023b). According to an
estimation based on a survey by Robra and Mastrullo (2011), 2.3% of the land, including meadows and pastures, is irrigated,
mainly with water directly from the Broye River. The most water-intensive crops are potatoes (50% of total regional irrigation
water use), maize (15%), tobacco (15%), and sugar beet (8%) (**?**). As shown in Figure 1, most irrigated crops are located in the
northern part of the catchment. We focus our modeling framework on potato fields because they have the most commercially
relevant demand for irrigation, accounting for around 50% of the water use in the area. Furthermore, reference data on irrigation
practices and bans are available, improving model validation and further analysis.

## 3   Data and methods

### 3.1   Methods

We use the agro-hydrological model SWAP (version 4.01), which simulates potato development and yield formation in response
to daily temperature and soil water availability to the plant. We apply the field-scale model at the catchment scale, using
high-resolution gridded soil data from the Swiss Competence Centre for Soil (KOBO) (2023) and gridded climate data from
MeteoSwiss (2021b, a) and Stöckli (2013). Irrigated potato fields are displayed with the soildata resolution of 30 m × 30
m, which corresponds to the resolution with which the model is run (see figure 1). The aim is to assess the effectiveness
of management measures in reducing irrigation demand and yield losses. This involves evaluating how seasonal irrigation
demand, crop yield, and drought stress change in response to irrigation bans and adapted management practices.

### 3.2   The soil-water-atmosphere-plant model SWAP

The soil-water-atmosphere-plant model SWAP is an open source, field-scale, 1-dimensional, vertically oriented, and physically-
based agro-hydrological model (Kroes et al., 2000; van Dam et al., 2008). The model uses a set of equations, such as the
Richard's equation, to simulate soil water flow, heat flow, and solute transport within the vadose zone (Noory et al., 2011). The
model requires daily climate input data and soil textural- and hydraulic parameters. According to Bonfante et al. (2010), SWAP
is superior to similar crop models (MACRO, Jarvis (1994), CropSyst, Stöckle et al. (2003)) in simulating surface infiltration,
drying processes and crop growth. This advantage is thought to be due to the fine resolution of the Richards' equation and
its numerical solution, particularly near the upper and lower boundaries (Bonfante et al., 2010; van Dam et al., 2008). SWAP
can be coupled with the WOrld-FOod-STudies (WOFOST) model (Supit and Van Diepen, 1994) to simulate detailed crop
growth based on light interception and $CO_2$ assimilation (Hu et al., 2019). In WOFOST, dry matter yield results from a gradual
reduction of the photosynthetically produced carbohydrates (=biomass) and the distribution of this biomass to different plant
organs, one of which is the potato tuber (Kroes et al., 2017).





**Figure 3.** Methodological framework

ten Den et al. (2022) introduced a more dynamic concept for the allocation of biomass to different plant organs. Instead of
using a tabular form for the allocation at different phenological development stages, we implement the sigmoid functions that
describe the share of biomass allocated to each organ depending on the plant's development stage, introduced by ten Den et al.
(2022). These functions allow a more smooth and realistic representation of biomass allocation (ten Den et al., 2022).



### 3.3 Data

#### 3.3.1 Meteorological data

As meteorological input, SWAP requires daily time series of minimum and maximum temperature, solar radiation, and precipitation. We use the corresponding data from the Swiss national weather service, MeteoSwiss, available on a 1 km × 1 km resolution grid (MeteoSwiss, 2021a, b; Stöckli, 2013)). In the lack of data on vapor pressure and wind speed for estimating evapotranspiration using Penmann-Monteith, SWAP uses the reference evapotranspiration (ET0) calculated with the Priestley-Taylor approach (Taylor and Priestley, 1972).

#### 195 3.3.2 Soil data

The digital map for soil properties by the Swiss Competence Centre for Soil (2023) provides soil textural and geochemical data for Switzerland at a resolution of 30 m × 30 m over three depths. Starting from points with measured data on soil parameters, the Swiss Competence Centre for Soil spatially interpolated soil properties using a quantile regression forest model and several covariates, such as climate and land use. The maps provide data on soil texture and organic carbon at depths 0–30 cm, 30–60

cm, and 60–120 cm (Swiss Competence Centre for Soil (KOBO), 2023). To estimate the Mualem-van-Genuchten parameters (Genuchten, 1980) required by SWAP, we used the pedotransfer function sets of the euptf2 package in R (Szabo et al., 2021), with the option "ptf02", that uses soil texture and soil organic carbon content as inputs.

#### 3.3.3 Observational data for model calibration and validation

We use two different datasets for SWAP model calibration: i) a field-scale dataset on irrigation water and ii) region-average

yield data without information on irrigation practices. The field-scale data is available from the Bern University of Applied Sciences, HAFL (School of Agriculture, Forest and Food Sciences HAFL, 2022). The field-scale dataset comprises information for a total of 33 station-years, where each station-year represents data collected from one potato field over one year. This dataset includes detailed data such as sowing and harvest dates, yield, soil texture, irrigation timing and amounts between the years 2018 to 2021. Note that not all fields have data available for each year within this range. Regionally averaged yield data are

obtained from a national survey of selected potato production sites within a 15 km radius of Payerne, the location of the nearest weather station (Agroscope, 2023). Subsequently, the data is aggregated to compute an annual mean from 1991 to 2022. Due to limited amounts of data, all reference information within the study area is used for calibration. We employed two datasets for model validation: (i) a field-scale irrigation water use dataset encompassing 61 station-years across the entire Swiss Plateau (School of Agricultural, Forest and Food Sciences HAFL, 2021), and (ii) region-average yield data for a 15 km radius

surrounding the Bern (BER) weather station situated within an agricultural area near our study region (Agroscope, 2023).




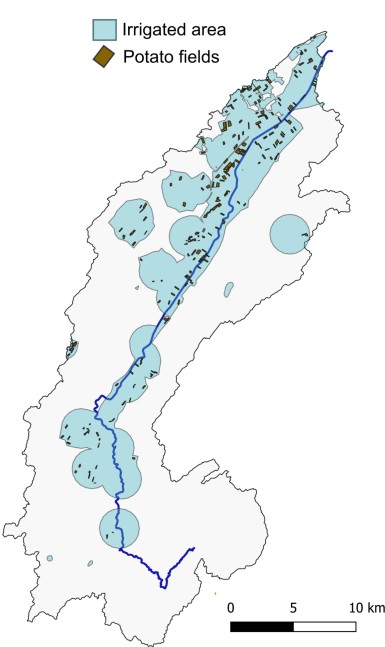

**Figure 4.** Potentially irrigated areas in light blue, (irrigated) potato fields within the perimeter for 2022 in brown (KGK-CGC, 2023).

## 3.4 Model set up

### 3.4.1 Model perimeter

To address the challenge of identifying the regularly irrigated areas, we estimate the irrigation perimeter based on technical considerations. In Swiss agriculture, it is reasonable to assume that mobile pumping stations transport water for an average distance of around 1600 me, contingent upon variables such as slope, hose diameter, and pump capacity (personal communication, Simon Baumgartner, smart farming engineer, 07.07.2023). Based on this rough estimate, we compute a buffer around all known water withdrawal concessions for (mobile) pumping stations (School of Agricultural, Forest and Food Sciences HAFL, 2023). We then adjust them based on logical constraints, such as tarred roads, and combined them with information on irrigated areas available from interviews in the region (Schaffner and Mastrullo, 2013). The resulting irrigation perimeter used for the modeling in this work is shown in Figure 4, including the location of irrigated potato fields (from very detailed field-scale land-use maps, KGK-CGC (2023). The vector data on potato fields is combined with fine-resolution grid cells of the soil map (resolution 30 m × 30 m) by assigning relative proportions of coverage by the potato fields to each grid cell (number between 0 and 1).



### 3.4.2   Other model set up options

We define the sowing date using the average sowing date from the field-scale reference data, which is the 15[th] of April, allowing the model to harvest whenever maturity is reached. In SWAP, we simulate irrigation by setting a period when it is allowed, the type of irrigation, a threshold to trigger irrigation based on soil properties, and the amount of water to apply when the threshold is reached. We set the irrigation period in such a way that 90% of the observed irrigation amounts from the reference data fall within this period. As a result, the irrigation period is set to 04[th] of June to 05[th] of August. This period is consistent with the

reference data and established irrigation practices (Fricke and Riedel, 2019; Kaspar et al., 2020). Irrigation is triggered by a predefined value of 40% depletion of plant available water (water held in the soil between field capacity and wilting point). We set the goal for the irrigation to bring the soil back to field capacity but define lower (10 mm) and upper amounts (30 mm) to represent common agricultural practice and the capacity of the installed systems. The irrigation type is set to surface irrigation. The bottom boundary condition of the model is set to free drainage. In 2022, several irrigation bans are imposed on water

withdrawal from the Broye and its tributaries. These bans, imposed between 23.06.2022 and 26.09.2022, either prohibited irrigation altogether or allowed it only at night for certain users (Châtelain, 2023). Consequently, in SWAP, the irrigation period is restricted to 05.06-23.06.2022.

### 3.5   Model calibration

Complex agro-hydrological models such as SWAP often include numerous parameters, leading to overparameterization and

prolonging the calibration process (Xu et al., 2016). A global sensitivity analysis (GSA) is performed to identify the most sensitive parameters for optimization. The crop parameters with the highest impact on yield and irrigation demand sensitivity are then optimized using a genetic algorithm (DEoptim, Mullen et al. (2011)) to closely match the reference data for the regionally-averaged annual yield and field-scale seasonal irrigation demand.

### 3.5.1   Global Sensitivity Analysis, GSA

We use Sobol indices (Puy et al., 2022), which have been used in several studies with SWAP to determine the most sensitive model parameters (Stahn et al., 2017; Wesseling et al., 2020; Xu et al., 2016)) to a certain model output. Sobol indices allow for a variance-based analysis of the first and total order of effects, meaning independent parameter effects and parameter interactions (Puy et al., 2022). We set the algorithm parameters $N$ (number of samples) and $R$ (number of bootstrap replicates) to 3000 and 200, resulting in 93000 model runs. To compute the sensitivity indices, we use Latin hypercube sampling (Wesseling

et al., 2020; Xu et al., 2016) as implemented in the sensobol R-package (Puy et al., 2022) to generate 3000 parameter sets. Guided by recommendations from the WOFOST manual (de Wit and Boogaard, 2021), we examine 29 parameters, including parameters that partially determine the phenology, $CO_2$ assimilation, root architecture, oxygen stress, drought stress, and biomass partitioning to different parts of the plant. The sensitivity of these parameters is tested to yearly crop yields and irrigation amounts. Each parameter is varied within the range of $\pm$ 15% of the default values for almost all parameters, as the default

parameterization already gives a good fit to the reference data (ranges shown in the Supplementary Material, Table S1). The





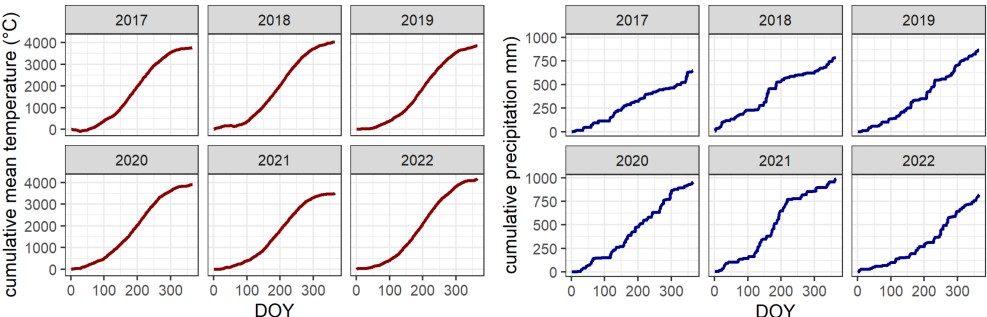

**Figure 5.** Cumulative mean temperature and precipitation in Payerne 2017-2022 (MeteoSwiss)

phenology of the crop is mainly determined by temperature sums in the form of growing degree days (GDD), which in SWAP mark different development stages (DVS) of the crop (0=sowing, 1=anthesis, 2=maturity). The different temperature sums are called TSUMOPT (growing degree days from sowing to emergence), TSUMEA (from emergence to anthesis) and TSUMAM (from anthesis to maturity). As their high relevance for yield simulation is well known (Utset et al., 2007), we exclude them from the sensitivity analysis (and use the default values for potato), but include them later in the parameter optimization.

Furthermore, a representative soil texture of the field-scale reference sites is used to run model simulations from 2017 to 2019. This period was chosen to capture meteorological variability. 2017 is characterized by relatively cool but dry conditions, 2018 by warmer temperatures and relatively dry conditions, and 2019 by temperatures in between the other years but wetter conditions than the other years (Figure 5).

### 3.5.2 Parameter optimization

We use the R package DEoptim (Mullen et al., 2011) to optimize the most sensitive parameters identified with the GSA. DEoptim employs a differential evolution algorithm (genetic algorithm) to develop different generations of parameter sets and shift them from the initial population towards a global optimum determined by the objective function defined by the user. Here, the following objective function ($F_{\mathrm{obj}}$) is used:

$$F_{\mathrm{obj}} = 2 - d_{\mathrm{yield}} - d_{\mathrm{irr}}, \tag{1}$$

where $d$ is the index of agreement defined (Willmott, 1981) as:

$$d = 1 - N\frac{M_{\mathrm{err}}}{P_{\mathrm{err}}}, \tag{2}$$

where $N$ is the number of observations, $M_{err}$ is the mean squared error, and $P_{err}$ is the potential error. Ideally, both $d_{\mathrm{yield}}$ ($d$ between observed and simulated yield) and $d_{\mathrm{irr}}$ ($d$ observed and simulated irrigation amount) are 1. The index of agreement is used to describe the model prediction error in a standardized way between 0 and 1, where 0 describes no agreement, and 1





is a perfect model prediction (Willmott, 1981). We aim for the sum of both to be as close to 2 as possible, thereby minimizing the objective function and approaching an optimal model configuration. We set the algorithmic parameters $N_P$ (number of generations) to 100 and the maximum number of iterations to 110, resulting in 11110 model runs. The parameter values of the initial population are set to the default SWAP values, and the search range for each parameter is again set to $\pm15\%$. We

determine TSUMEA and TSUMAM based on observed sowing and harvest dates. Thereby, the total growing degree days, representing the sum of daily temperatures exceeding a 2°C baseline during the growing seasons, is calculated using field-scale data. Note that below 2°C, there is no potato growth, (Kroes et al., 2017). The growing season is defined as the period between sowing and harvesting dates, resulting in an average of 2149 degree days over all station-years. TSUMEA and TSUMAM are then calibrated to the observed data, TSUMOPT is left at its default value of 170°C and is subtracted from the total 2149°C.

From the resulting 1979°C, the parameter TSUMEA, which is determined during optimization, is subtracted, which in turn gives the value for TSUMAM for each parameter set.

### 3.6   Soil and crop management scenarios

Two different management scenarios on irrigation demand (and, subsequently, yield) are evaluated with SWAP: an increase in soil organic carbon (SOC) and the adoption of earlier maturing varieties. We study the effects individually and combined, as

well as with and without considering irrigation bans, resulting in 20 management scenarios (Table 1). In line with Turek et al. (2023), we assume that adaptations in soil management (e.g., organic amendments, cover cropping) lead to an accumulation of SOC of an additional 1% down to a depth of 60 cm. An increase in SOC to varying degrees due to different management has been documented in many publications (Diacono and Montemurro, 2010; Gross and Glaser, 2021; Lianhai, 2022). We select a high value of +1% to analyze the upper limit of the potential with this somewhat hypothetical scenario. The crop file calibrated

and used in our study for baseline conditions represents the medium-late maturing varieties Innovator or Agria, which, on average, have a 140-day growing season in the Broye catchment. In contrast, common Swiss early to intermediate maturing varieties like Fontane, Ivory Russet, Agata, Lady Christl, or Annabelle require as little as 90 days to reach maturity (Schwärzel et al., 2022). To simulate these earlier maturing varieties, we adjust the phenology by recalculating the average growing-season growing degree-days for shortened growing periods (130, 120 and 110 days) based on the observed temperature sums from all

station-years from the reference field-scale data. To further explore the effect of the growing season length, we also include an even later maturing variety with a growing season length of about 150 days.

### 3.6.1   Representative subsample

Based on the above management options, we simulate the 20 scenarios (see Table 1) for a representative subsample of potato fields of the study area. This way, we can pre-select the most promising scenarios for the regional application (which are

computationally demanding). We randomly sample 50 cells (a field is composed of several cells) to obtain a representative subsample and apply each scenario for the period from 2019-2022, focusing on the results from 2022. The most promising scenarios for the regional analysis are then selected based on several criteria: Irrigation Water Productivity (IWP), irrigation demand reduction, yield loss level and transpiration gain. IWP is computed as the ratio of total crop yield per cell (in deci-tons,



**Table 1.** Scenarios of management combinations applied to the representative subsample

| Maturity (growing season length [d]) | Scenario | Irrigation bans | Increased SOC content (+1%) |
|---|---|---|---|
| (default) late-mid maturity (140 days) | 1 | x | |
| | 2 | x | |
| | 3 | x | x |
| | 4 | x | x |
| mid maturity (130 days) | 5 | x | |
| | 6 | x | |
| | 7 | x | x |
| | 8 | x | x |
| early-mid maturity (120 days) | 9 | x | |
| | 10 | x | |
| | 11 | x | x |
| | 12 | x | x |
| early maturity (110 days) | 13 | x | |
| | 14 | x | |
| | 15 | x | x |
| | 16 | x | x |
| late maturity (150 days) | 17 | x | |
| | 18 | x | |
| | 19 | x | x |
| | 20 | x | x |

dt) divided by the total irrigation water amount ($m^3$). The transpiration gain gives a measure of drought stress, in which a high

transpiration gain means a lower drought stress-induced transpiration reduction relative to the reference scenario (scenario 1 of Table 1).

We consider a scenario relevant for regional analysis if it yields a high IWP and transpiration gain and decreases irrigation demands without significantly compromising yield.

### 3.6.2 Regional application

After selecting the most relevant management scenarios from the representative subsample (Section 3.6.1), we apply the model under the most relevant scenarios to the entire study area (10129 grid cells) for the period 2019 to 2022. The warm-up period is set to three years (required for state variable initialization; this data portion is discarded before analyzing the results). SWAP





is run for each grid cell using the corresponding meteorological and soil data. The resulting outcomes (irrigation demand in l
$m^{-2}$ and yield in kg ha$^{-1}$) are summed over the irrigated fraction of each cell (see section 3.4.1). The total computation time
to run the model for 4 years and the entire study area on a PC with 16GB RAM is 17 hours.

### 3.6.3 Optimal spatial configuration of management options

The management scenarios shown in Table 1 are homogeneously applied to all grid cells. Based on which of these scenario
yielded the highest IWP for each cell, this scenario is retained as the best management choice for that cell. We then run the
model for each grid cell using this management to evaluate the impact of locally adapted management choices on regional
irrigation demand and yield. This model run is referred to as the "best scenario".

## 4 Results

### 4.1 Model calibration and validation

The global sensitivity analysis (GSA) revealed 8 parameters that, additionally to the temperature sums (TSUMEA and
TSUMAM), exhibit significant sensitivity concerning yield and irrigation amount (default values are shown in Table 2, full
GSA results in Supplementary Material, Figure S1). This sub-selection of 8 out of the 29 crop parameters tested in the GSA is
unsurprising, as they are either strongly related to crop yield or drought stress. SLATB represents the specific leaf area (which
influences light interception) and AMAXTB represents the maximum $CO^2$ assimilation rate, depending on the development
stage (DVS). Both light interception and $CO^2$ assimilation are major drivers of crop growth and, thus, yield. As a function of
crop water dynamics, the maximum rooting depth (RDC) of the crop plays an important role in the required irrigation water
demand. ALPHACRIT and ADCRL are parameters of the Feddes-Jarvis function of root water uptake reduction (Feddes et al.,
1978). ADCRL denotes the pressure head above which root water uptake will be reduced due to drought at low atmospheric
demand. Low atmospheric demand is defined here by lower temperatures and evaporation. ALPHACRIT is a critical stress
index that indicates the transpiration rate at which the plant can compensate the water uptake reduction due to drought stress.
It is assumed that the plant can compensate low root water uptake from drier parts of the soil with (still) wetter parts in the soil
(Jarvis, 2011).

The remaining sensitive parameters, FLTBb, FLTBc and FOTBc, relate to the partitioning of the generated biomass into
different plant organs, which also strongly influences yield. The corresponding partitioning into the storage organs (FOTB),
leaves (FLTB) and stem (FSTB) is given in the Supplementary Material, Table S2. If a value is 0, no biomass is partitioned to
this storage organ at this DVS.
The optimization of relevant model parameters identified in the GSA enabled an improved fit of the simulated to the observed
data (Figure 6). The fit of simulated to observed field-scale irrigation amounts increases from $d$=0.54 to $d$=0.84 (for optimized
parameters), and the fit of simulated to observed potato yields from $r$=0.58 to $r$=0.71 (optimized parameters) for the field-scale
simulation of irrigation amounts. The fit to the region-averaged reference data on yield increased from $d$=0.42 to $d$=0.67 and





**Table 2.** Default and optimized parameter values depending on development stage (DVS)

| Parameter | Definition | Unit | Default | Optimized | DVS |
| --- | --- | --- | --- | --- | --- |
| TSUMEA | temperature sum from emergence to anthesis | °C | 150 | 166 | |
| TSUMAM | temperature sum from anthesis to maturity | °C | 1550 | 1813 | |
| CF | crop adjustment factor for reference Evapotranspiration | | 1 | 0.87 | 0 |
| CF | crop adjustment factor for reference Evapotranspiration | | 1.10 | 0.96 | 1 |
| SLATB | specific leaf area as function of DVS | ha kg$^{-1}$ | 0.0030 | 0.0033 | 0 |
| SLATB | specific leaf area as function of DVS | | 0.0030 | 0.0033 | 1.1 |
| SLATB | specific leaf area as function of DVS | | 0.00150 | 0.00165 | 2 |
| ADCRL | level of low atmospheric demand | cm d$^{-1}$ | 1 | 0.087 | |
| RDC | maximum rooting depth | cm | 50 | 43.85 | |
| ALPHACRIT | Critical stress index for compensation of root water uptake | | 1 | 0.998 | |
| AMAXTB | max CO2 assimilation rate as function of DVS | kg ha$^{-1}$ h$^{-1}$ | 30 | 33.89 | |
| FLTBc | regulates fraction of dry matter partitioning function to leaves at DVS=0 | | 0.83 | 0.95 | |
| FOTBc | regulates fraction of dry matter partitioning function to storage organs at DVS=2 | | 1 | 1.00008 | |
| FLTBb | regulates the location of the center point of dry matter partitioning function to leaves | | 1.05 | 0.91 | |

from $r$=0.46 to $r$=0.61. The fit also improved when the model was validated with reference data from outside the study area, indicating transferability to other regions and no over-fitting of parameters (Figure 6).

## 4.2 Impacts of drought and supply deficit on crop productivity

For the year 2022, the overall regional irrigation demand was simulated to be 697'389 m$^3$. The amount of water actually supplied (considering the temporal irrigation bans) was simulated to be 285'836 m$^3$. This means that only 41% of the demand could be satisfied, i.e., an irrigation deficit of 59%. In terms of quantity, this translates into a yield deficit of about 16.4%.

To explain what drives the spatial variability in mean seasonal irrigation demands (Figure 7), we conducted a principal component analysis (PCA, full results in A1). We used the mean irrigation amounts in l/m$^2$ for each of the 10129 cells as the dependent variable; as predictors, we included edaphic and climatic variables and the growing season length. As expected, higher seasonal irrigation amounts are associated with higher sand contents and with higher bulk density. In contrast, lower irrigation amounts are associated with higher silt and clay contents, with higher SOC contents and with higher residual and





**Figure 6.** Calibration: Comparison of simulated and observed field-scale seasonal irrigation amounts (33 station-years) and regional-scale yield data (15 km around Payerne), showcasing default and optimized parameter values. Validation: Extending the comparison to 61 station-years across Switzerland for seasonal irrigation amounts. Regional-scale yield data (15 km around Bern) evaluated with default and optimized parameters.)





saturated water contents of the soil. These results, therefore, verify the plausibility of the spatial variability we observe. The spatial heterogeneity of soil properties leads to significant differences in the amount of irrigation within agricultural fields (Figure 7). Consequently, the most efficient management scenario also varies within agricultural fields.

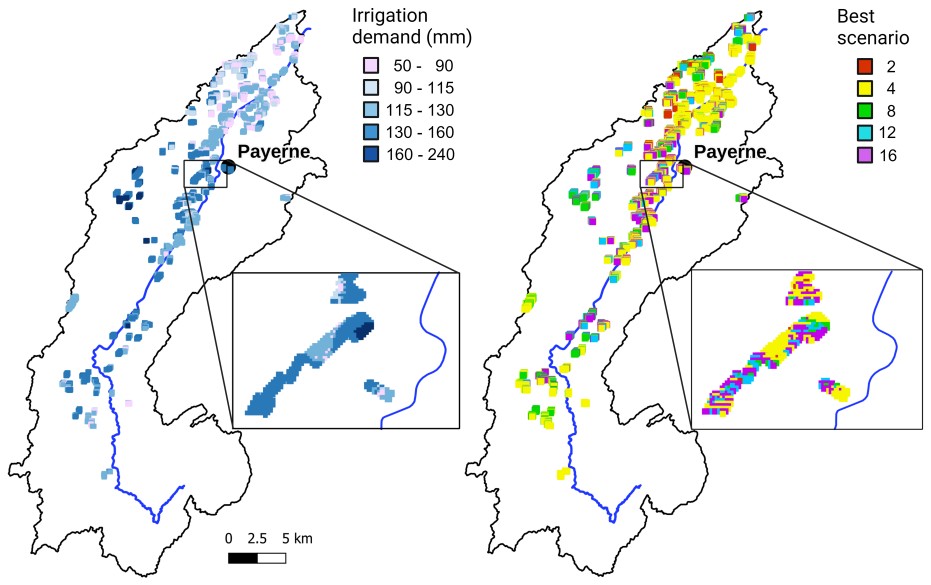

**Figure 7.** Left: seasonal irrigation demand per cell in 2022 without considering irrigation bans or management (reference scenario 1). Right: Best management scenario per cell considering the highest irrigation water productivity.

## 4.3 Adaptation scenarios

### 4.3.1 Pre-selection of scenarios

Based on the full results of the scenario analysis for the sample of 50 randomly sampled cells (see Supplementary Material, Table S3), we identified scenarios 4, 8, 12 and 16 as most promising. While scenario 4 considers increased SOC content and irrigation bans, in scenarios 8, 12 and 16, increasingly earlier maturing varieties are used. We are mainly interested in how the management choice impacts agricultural productivity when considering irrigation bans. Therefore, we excluded scenarios with low IWP, low transpiration gain, and with too high reduction in yield or too low reduction in irrigation demand. Since a

change in agricultural practice should not lead to a higher water demand (i.e., a reinforcement of water-use conflicts), we had to exclude the scenarios with the late-maturing variety (scenarios 17–20). We also excluded the scenarios that only considered earlier maturing varieties (scenarios 6, 10, 14) because these scenarios result in notable yield losses. The scenarios that most reduce the irrigation demand are those with earlier maturing varieties (16, 12 and 8). In these scenarios, the yield reduction can partially be compensated by the SOC increase. Although scenario 16 still leads to a relatively high yield reduction, the high

transpiration gain indicates low drought stress. We decided to consider scenario 4 as it retains a high IWP through a substantial increase in yield, even though irrigation demand could not be decreased in this subsample.







**Figure 8.** Results for the simulations applied to all 10129 cells under the reference and the selected management scenarios, as well as the "best scenario", comprising the results for adjusted management to maximize IWP, further explained in section 4.4.

### 4.3.2 Regional irrigation demand and deficits under selected management scenarios





Figure 8 shows the results for the simulation of all irrigated potato fields in the Broye catchment in 2022 for the promising scenarios selected in section 4.3.1. The results for each scenario are compared to the reference scenarios. Our simulations
identified scenario 4 (increased SOC) as having the highest total yield. Conversely, scenario 8 exhibited the greatest amount of "realized" total irrigation (the fulfilled demand during water restrictions). Additionally, scenario 4 displayed the highest level of drought-induced transpiration reduction, indicating the most severe drought stress, while the "best scenario" resulted in the least.

It can be observed that the regional results did not completely mirror the results of the subsample described in Section
4.3.1: On the subsample level, the scenario with the highest irrigation amount was scenario 4; on the regional level, scenario 8, shows the highest total irrigation amount, but not the highest yield, which is lower than for scenario 4. Accordingly, the IWP of scenario 8 is the lowest within this group of scenarios. The highest yield is obtained with scenario 4, which has a similar total irrigation amount as scenario 12 (earlier maturing variety), but the highest IWP. The highest drought-induced transpiration reduction (again interpreted as higher drought stress) is also obtained for scenario 4. The lowest transpiration
reduction(i.e., lowest stress level) is obtained for scenario 16, featuring the earliest maturing variety. In terms of compensating yield loss, scenario 4 appears to be the best option on the regional scale, with a slight decrease in irrigation demand (in the case of irrigation bans) and a considerable increase in yield.

## 4.4 Site-Specific management impact on regional irrigation demand

In Figure 8, we show the results for the scenarios where we homogeneously applied the same management to the whole
study area. The "best scenario" column showcases the simulation results where, for each individual grid cell, the scenario with the highest Irrigation Water Productivity (IWP) was chosen and applied. Under this "best scenario" case, irrigation water demand relative to the reference scenario (2) is decreased by 26% and yield is increased by 2%. The IWP is the highest over all scenarios. When not considering irrigation bans, irrigation demand is still decreased by 9% and yield increased by 2% (see Supplementary Material, Table S4). We conducted another PCA to illustrate under which combination of pedoclimatic
conditions which management applications are most efficient. We omitted the growing season length as a predictor variable here, as this is changed in most scenarios.

In the resulting Biplot (Figure 9), we can identify a general trend, although the groups cannot be clearly separated. As expected from the results in Figure 8, scenario 4 had the highest IWP in most locations and, therefore, also dominated this visual representation. Most cells where scenario 4 had the highest IWP were associated with low irrigation levels and high
clay and silt content (i.e., favorable edaphic conditions, especially on the right side of the y-axis). In exchange, in the cells with high irrigation amounts and high sand content, the scenarios with earlier maturing varieties were more prominent than other scenarios. This indicates that an increase in SOC alone (scenario 4) is sufficient to promote high IWP values for cells with favorable edaphic conditions (high clay and silt content). For cells with unfavorable edaphic conditions (low silt and clay content, high conductivity or high bulk density), SOC increase is often insufficient to promote high IWP values; these locations
would instead additionally require earlier maturing varieties for more efficient water use.



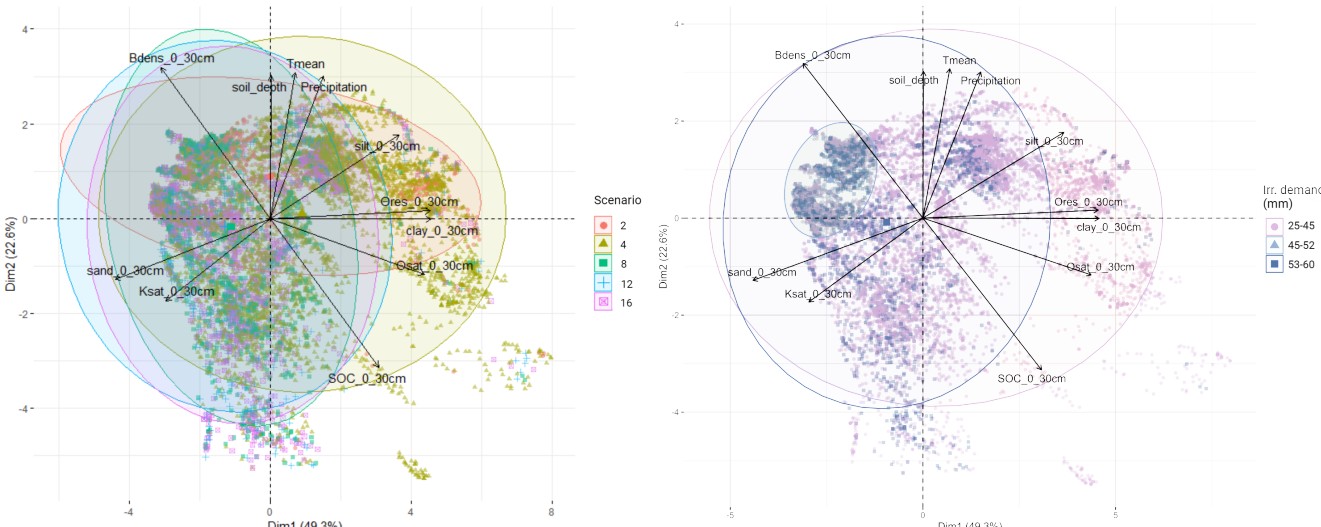

**Figure 9.** Left: Biplot showing the observations and predictor variables within the dimensions of the first two principal components. Observations show the seasonal irrigation amount for 2022 without considering irrigation bans or management. Color coding corresponds to the best management scenario per cell, based on IWP (only the ones considering irrigation bans, i.e., 2,4,8,12 and 16). Right: Same biplot, but color-coding shows the seasonal irrigation water amount simulated under the associated best management scenario. Predictor variables are further explained in Appendix A1

## 5 Discussion

### 5.1 Potentials and limitations of the approach

The modelling approach employed in this study provided plausible estimates of yield deficits that could be attributed to irrigation water resource limitations during a recent drought year. It should be noted, however, that the drought pattern in 2022 (timing, duration and magnitude) might differ from other years as will the effect of irrigation bans and adaptive management. However, the results of this case-based evaluation have a high value for communication (Sprain and Timpson, 2012) to the broader public and regional stakeholders as it connects closely to personal experiences.

Estimating the regional irrigation demand requires detailed input and reference data, which might often not be available. In this study, especially the extent and location of the potentially irrigated fields are a rough estimation and include some uncertainty. However, where observations or estimates of irrigated areas are available, our approach could be used to fill important knowledge gaps on regional irrigation water demand. When coupled with climate projection data, estimates of future irrigation demand could inform the planning of irrigation infrastructure. Other options for climate adaptation, such as mulching, cover cropping, or reduced soil tillage, should also be taken into account.

The model was applied on a 30 m resolution grid, which revealed considerable heterogeneity in irrigation demands across agriculture fields (see Figure 7). However, the framework could also be applied at larger scales, such as the actual field scale,



to reduce computational costs. This may be particularly relevant in larger catchments or where there is a higher proportion of irrigated cropland.

## 5.2 Impact of irrigation bans on agricultural productivity in the case study

We simulated the regional irrigation demand for potatoes within the Broye catchment in 2022 with and without considering
the temporal bans on water withdrawal for irrigation under different scenarios. The model results show a significant deficit in irrigation water supply of 59% and a subsequent yield deficit of 16.4% for the reference scenario 1 (without any management adaptation).

To put the yield reduction into perspective, we can refer to observed yields of potato in Switzerland for the year 2022, which were reduced by 11% relative to 2017-2021, and by 22% relative to the high-yielding year 2020 (Federal Office for Agriculture
(FOAG), 2023a). This decline is thought to be due to extreme drought and heat, leading to quality degradation that could not be avoided due to local bans on irrigation (Federal Office for Agriculture (FOAG), 2023a).

Regional estimations on irrigation demand are generally challenging to validate, primarily due to a lack of data on demand or estimates on deficits in irrigation water supply, as well as the resulting deficits in harvest. In places such as Switzerland, where water scarcity has rarely been an issue in the past, this data is even more rare. As droughts are projected to increase in
magnitude and frequency in Switzerland (CH2018, 2018), the associated upcoming information demand must be addressed. Datasets that cover information on how much water would theoretically be needed and how much water is likely to be lacking in a drought year are essential tools to support the implementation of water retention or harvesting measures. The year 2022 was the second-warmest year in the study region since the weather station in Payerne started operating in 1964, only beaten by the year 2023 (MeteoSwiss, 2024). The high levels of irrigation deficit and yield reduction simulated for 2022 are expected
to be a re-occurring phenomenon. Since our estimations account only for quantity, it is impossible to estimate the marketable yield share from the total harvested yield for our study region. However, we can infer that the scenarios that display substantial drought stress will probably have a further reduction in yield through an increase in the unmarketable produce. A decrease in transpiration, often a result of stomatal closure, is a response of crops to heat and drought stress that conserves water but reduces photosynthetic assimilation, meaning yield formation (Obidiegwu et al., 2015). Due to their relatively weak and shallow root
systems, potatoes are highly susceptible to drought, especially in combination with soil compaction (King et al., 2020). The reduction in yield quantity, for example, through a reduction in the number of tubers, is therefore apparent (Nasir and Toth, 2022). But also the quality of the harvestable yield can be impacted, as drought and heat stress may lead to physiological defects of the tubers, leading to relatively small (<3cm) unmarketable tuber (Nasir and Toth, 2022), or leaving them deformed or immature at harvest (Obidiegwu et al., 2015; Rykaczewska, 2017). The extent of these impacts is highly dependent on the
timing and duration of the drought event and the potato variety (Obidiegwu et al., 2015; Rykaczewska, 2017). Drought stress during tuber onset often impacts quality, while drought stress during tuber bulking rather affects quantity (King et al., 2020).




### 5.3 Potential of soil and crop management adaptations

By increasing SOC over all cells (scenario 4), yield deficits due to irrigation bans can be reduced from -16.4% down to -7%. Even if the irrigation demand is slightly reduced, the transpiration reduction due to drought stress is higher than in reference scenario 2. We further observed that with the increase in SOC and related amplified plant growth, the regional irrigation demand was slightly increased when not considering bans (scenario 3). When considering irrigation bans (scenario 4), the demand was only slightly reduced, meaning that the effect of improved water retention at this level of SOC increase can be overpowered or reduced by the increased demand to sustain the growth.

The enforcing effect of increased SOC on water retention is well studied (Diacono and Montemurro, 2010; Eden et al., 2017), as is the effect on crop yields we observed. Porter et al. (1999) evaluated the effect of enhancing SOC levels on potato yield and irrigation demand. During field trials in Maine in 1993-1995, organic matter content was increased by +1.2% due to cover cropping and organic amendments. They found while enhancing SOC alone did not make up for a lack of irrigation, potato yields could be improved significantly. While the increase in SOC is a long-term goal of applying organic amendments, even one-time applications may already have positive effects on the share of marketable potato yields (Rittl et al., 2022). As also observed in our results, a general increase in SOC is useful to reduce yield losses. Still, it cannot suppress the irrigation demand or fully mitigate drought stress.

When the increase in SOC was combined with earlier maturing varieties, we observed only a marginal reduction (scenario 12) or even an increase (scenario 8) in the regional irrigation demand. One explanation is that the earlier maturing varieties have higher transpiration levels earlier on, which increases root water uptake at this stage. If this demand cannot be met by precipitation, irrigation demand increases. Due to the irrigation bans, irrigation in 2022 is only possible in a small window of time. If this window now coincides with the period where transpiration and root water uptake of the earlier maturing varieties are higher, their irrigation demand will be higher, too. When this stage coincides with the relatively small window in which irrigation is possible, given irrigation bans, the demand for the earlier variety may be greater than for the later variety. In addition, the earlier maturing varieties tested here develop a shallower root system, which can make the plant more dependent on additional irrigation at critical stages of development. The IWP for the scenarios with earlier maturing varieties is, however, relatively low, as those have less time to develop tubers. The ability of different varieties to escape or combat drought impacts relies a lot on the timing, magnitude and duration of the drought. This was observed by Chang et al. (2018), who analyzed the drought impacts on the canopy development and tuber growth for different maturity classes of potato.

We also analyzed under which conditions we find higher irrigation demands and under which conditions management leads to the highest productivity. As expected, our analysis confirmed that soil texture predominantly influences water retention capacity and plant water availability, consequently affecting the extent of the irrigation requirements. Soils with higher sand content or bulk density exhibit greater seasonal irrigation demand than those with higher clay and silt content. Increased precipitation during the growing season reduces irrigation needs. Higher temperatures result in shorter growing seasons due to their accelerating effect on phenology, thereby lowering seasonal irrigation demands. The spatial variation in seasonal irrigation demand observed in our study region emphasizes the need for site-specific water and crop and soil management.



We could also see that in locations with favorable pedoclimatic conditions, the scenario with increased SOC (4) leads to the highest IWP. In contrast, earlier maturing varieties (together with increased SOC) increase IWP in locations with less favorable conditions. Similar results were obtained by Ahmadi et al. (2010), who concluded that soil texture plays a significant role in choosing the best irrigation practice to maximize water productivity.

The scenario with management adapted to each site ('best scenario' in Figure 8) did indeed produce the greatest efficiency of irrigation water and reduction in irrigation water demand. This site-specific crop and soil management can reduce irrigation water demand by 26% while increasing yields by 2% compared to no management (scenario 2). In relation to scenario 1, the yield loss of -16.6% can thus be reduced to -14.4%. This approach is particularly valuable from the perspective of sustainable use of resources, as the reduction in yield loss is marginal. We observed a high IWP and transpiration gain. This effect is

retained when there are no water restrictions at all. The drought-induced transpiration reduction ($Tred_{dry}$) is used as a drought stress indicator in SWAP. A reduction of $Tred_{dry}$ compared to the reference scenario would result in a transpiration gain. The high transpiration gain here, therefore, indicates that the crop was less stressed (compared to scenario 2) and, therefore, less affected in terms of quantity (as visible by an increase in yield) but likely also quality.

### 5.4 Practical limitations to managing SOC stocks

Our assumption of an increase in SOC by +1% down to 60cm depth was supported by experimental studies which had shown that such differences could be achieved through particular types of soil management (e.g., cover cropping, continued applications of compost and other organic amendments, reduced tillage (Wezel et al., 2014; Diacono and Montemurro, 2010; Eden et al., 2017; Holland, 2004; Hou et al., 2012)). For example, Diacono and Montemurro (2010) reviewed long-term experiments on the organic amendment and found that continuous applications may lead to an increase in soil organic carbon by up to

90%. In a comprehensive meta-analysis, Gross and Glaser (2021) found an average increase by 35% of SOC stocks following organic amendments. Depending on the initial SOC content, this increase would result in additional 0.25% (initial SOC=0.5%) to 1.13 (initial SOC=2.5%). The highest relative increases (48%) were found in soils with <1% initial SOC content and higher clay content. SOC levels in our study region are 1-2%; most soils can be classified as loam, clay loam or sandy loam, so we expect the potential increases to be closer to the average of 35%. Porter et al. (1999) reported an increase in organic matter by

+1.2% that roughly corresponds to an increase in SOC by +0.7%.

In conclusion, the static increase of +1% SOC can in principle be achieved. However, it remains to be investigated in future studies what types of management adaptations could lead to the SOC increase assumed in this study and whether it would be economically viable (given the regional pedo-climatic conditions in the case study region). The level of SOC increase applied here requires significant sources of organic material, such as compost or manure, which may not always be available due to the

management system (low livestock production) or conflicts of use (e.g. biogas production). Efforts to increase the supply of organic matter would require long-term and systemic adaptations, as SOC stocks are expected to decrease due to accelerated decomposition with continuously increasing temperatures (Wiesmeier et al., 2016).

Irrespective of these considerations, our study emphasizes the critical role of soil organic carbon (SOC) in drought resilience, particularly as droughts are projected to intensify with climate change. Maintaining and enhancing SOC through soil manage-



ment not only benefits moisture retention but also offers the co-benefit of carbon sequestration, with significant potential in Switzerland (Keel et al., 2023).

## 6   Conclusion

This study presents a comprehensive evaluation of climate adaptation options in soil and crop management to mitigate yield losses and to increase irrigation water productivity during periods of water scarcity in a recent drought year. Our focus is
on the Broye River catchment in Switzerland, which is representative of similar mid-sized lowland catchments in Central Europe. Therefore, we can provide valuable insights into the challenges faced by regions experiencing reduced low flows and subsequent limitations in irrigation water supply from surface waters.

Our analysis indicates that irrigation bans and the summer drought in 2022 significantly reduced potato yields by 16%, attributed to a 59% deficit in irrigation water. Hypothetical adaptation scenarios suggest that adding 1% soil organic carbon
down to 60 cm depth could reduce the drought and irrigation ban-induced yield loss from -16.4% to only -7%. Additionally, planting earlier maturing potato varieties in sites with less favorable pedoclimatic conditions could enhance irrigation water productivity and decrease irrigation demand by 26%. Yield losses, in this case, could only be reduced to -14.8%. These results highlight the great value of soil organic carbon for preventing productivity losses during droughts and show that irrigation water use efficiency increases can best be promoted by location-specific combinations of adaptation choices. Alarmingly, SOC stocks
are observed and projected to decline in response to stagnating crop yields and climate change. This calls for the maintenance and enhancement of SOC through soil management that not only promotes adaptation in the form of increased soil moisture retention but also enhances mitigation through carbon sequestration.

The regional application of the field-scale, physically based SWAP model enabled simulations of regional irrigation demand and potato yields and deficits in response to drought stress. Such estimates are not only critical for planning irrigation or water
retention infrastructure but also highlight the need for proactive measures to reduce reliance on supplemental irrigation. Future studies should investigate the large-scale impact on a broader range of adjustments in crop- and soil management strategies, including conservation tillage, mulching and cultivating better-adapted crops. Additionally, the impacts of large-scale adoptions of adjusted management on the hydrological cycle at the catchment scale should be explored. This will contribute to developing more holistic and resilient agricultural systems capable of adapting to changing hydrological conditions in the face of increasing
drought extremes.

*Code availability.*   The code is available on GitHub (https://github.com/MalveHeinz/SWAP_regional)

*Data availability.*   See main text.




*Author contributions.* Conceptualization: MH, AH, BS; Methodology: MH, AH; Data curation: MH, AH, AK; Formal analysis: MH; Resources: MH, AH, BS; Supervision: AH, MT, BS; Writing-original draft: MH; writing-review & editing: MH, AH, BS, MT

*Competing interests.* The authors declare no competing interests.

*Disclaimer.* The authors do not have any competing interests. Irrigation water withdrawal locations as of 2022 according to HAFL survey. There is no claim to completeness. FOEN or HAFL assume no liability and give no guarantees for the correctness of the data.

*Acknowledgements.* The work was mainly funded by the Bretscher Fonds managed by the Oeschger Centre for Climate Change Research (OCCR) and the Federal Office for the Environment (FOEN) through the SwissIrrigationInfo project, as well as the EJP SoilX project by the European Union Horizon 2020 research and innovation programme (grant agreement no.862695). We would also like to thank Christoph Raible for reviewing and revising the manuscript. We used AI tools in polishing the language, namely deepL and Gemini. Several graphs were created with the help of BioRender.com.

**Appendix A**

**A1 Spatial variability of irrigation demand**



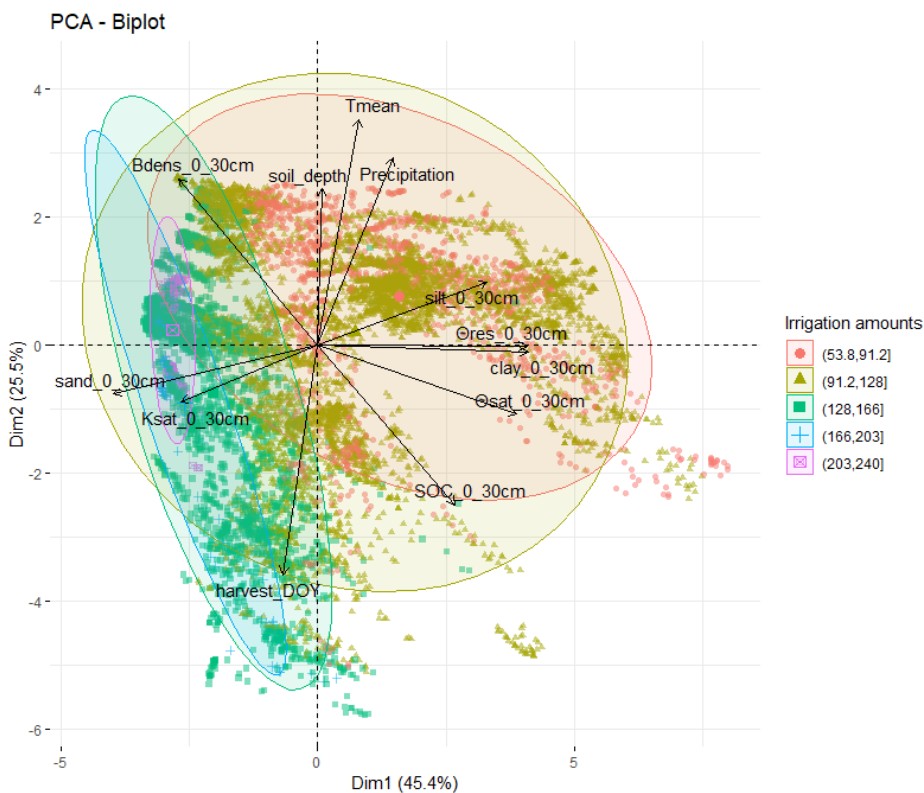

**Figure A1.** Biplot showing the observations and variables within the dimensions of the first two principal components that explain 72% of the total variance. Observations show the seasonal irrigation amount for 2022 without considering irrigation bans or management. Color coding corresponds to 5 equal ranges of irrigation amounts. Edaphic parameters are always taken from the first 30cm, Bdens = bulk density, soil_depth = total depth of the soil profile, Tmean = mean temperature over the growing season, Precipitation = cumulative precipitation over the growing season, Ksat = saturated hydraulic conductivity, harvest_DOY = day of the year at harvest, $\theta$res = residual water content, $\theta$sat = saturated water content



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
