# Peer review of "Can adaptations of crop and soil management prevent yield losses during water scarcity? - A modelling study"

_EGUsphere, 2024_

## Author Response (AR2)

**Suggestions for revision or reason for rejection by anonymous referee #3**

*"The authors have substantially revised the manuscript in the light of the review comments. However, the creditability of the model application and the consequent results could be improved by a rigorous calibration and validation of the soil hydraulic properties (e.g., soil moisture related parameters). For models like SWAP, a good calibration of soil hydrology is pivotal for simulation of the crop growth processes and any further scenario analysis. The current manuscript does not provide sufficient information on this aspect, which raises serious concerns about the reliability of this study. Nevertheless, the authors do mention at page 8 (section soil data) that, "To estimate the Mualem-van-Genuchten parameters (Genuchten, 1980) required by SWAP, we use the pedotransfer function sets of the euptf2 180 package in R (Szabo et al., 2021), with the option "ptf02", that uses soil texture and soil organic carbon content as inputs." This is a very basic approach, and lack scientific rigour in achieving a well functioning and justified model in the actual field conditions. Therefore, it is important to address this issue before a publication can be recommended. Hope authors can do a major revision on this or at least provide more information and discuss the limitations of their approach.*
*Another minor comment is to revisit the statement in introduction section (line 110),"To our knowledge, no study has simulated irrigation demand while considering restrictions on water supply and the impact of these on yield for temperate regions." To be on safe side, this statement can be deleted. Without this statement, the research gaps and study justification still remains high."*

**Reply to anonymous referee #3**

Thank you for the comments. To account for them, we revised the introduction, the section on soil data and the discussion section on limitations.

We clarify that for this regional model application, we decided to build on pedotransfer functions (PTF)-from the literature to estimate soil hydraulic properties due to limited spatial information on soil properties in our case study. Calibration of soil hydraulic properties, or building our own PTF, would require time series of soil moisture measurements, which at best would be available for a few individual sites within the catchment. Furthermore, if we had performed a calibration for single sites, this would hardly be transferable to other cells within the catchment where reference information for calibration is not available.

For this reason, the use of PTFs from the literature to derive soil hydraulic properties was the best feasible approach here and it is, in fact, a fairly common one, especially in spatialized applications. Weber et al. (2024) argue that if the spatio-temporal state of the soil is known, PTFs can indeed be used to estimate soil hydraulic properties from soil maps for further use in numerical models. We are aware of the scaling issue; PTFs are developed at the field scale and then (often) applied to regional or even global scales, which can lead to a scale mismatch (Weber et al. 2024). In our case, the relatively high resolution of the soil map (30m x 30m) still allows the representation of heterogeneous patterns. But of course, uncertainty remains and arises from applying this from point-measurements derived PTF to average soil properties over 90m$^3$. In this new version of the manuscript, we included the discussion of this issue (lines 430-438).

In the revised manuscript we further highlight the quality and value of the soil map by the Swiss Competence Centre for Soil (2023). The soil map is the product of a model trained and validated on over 30`000 measurements of soil properties in Switzerland. The results were validated and the sampling

density in our study region was high (due to intensive agricultural use), which gives us some confidence that the provided soil properties are well represented in the map. With this as a fairly solid input, we could feed the pedotransfer function from the euptf2 package, which was trained on over 8'000 soil samples across Europe and represent the state of the art for PTFs in Europe (Szabó et al. 2021). For ptf02 (used in this study), the $R^2$ of predicted and measured Mualem-van Genuchten parameters of both the training and test datasets is >0.8 (>400 samples were available for this specific PTF and prediction target). Considering only a Swiss case scenario, this exact PTF was applied and validated by Turek et al. (2023) on a typical Swiss agricultural soil. They found a median correlation of r= 0.79 between simulated and measured soil moisture at different depths. The quality of the soil map and the validation by Turek et al. (2023) therefore give us confidence in the choice of this particular approach, although it was not possible to calibrate or validate it ourselves. We list below where this information was added to the manuscript text:

1) Section 3.1.2, line 179ff:

*"The used soil map is the product of a model trained and validated with a series of predictor variables combined with over 30'000 measurements of soil properties in Switzerland, which results in more than 1 sample per km².*

*(…)*

*Although no reference data was available to validate our estimates, we built on the literature that suggests that if the spatio-temporal state of the soil is known, PTFs can indeed be used to estimate soil hydraulic properties from soil maps for further use in numerical models (Weber et al. 2024). Considering a typical Swiss agricultural soil, Turek et al. (2023) applied this specific pedotransfer function and validated it with soil moisture measurements (r = 0.79)."*

2) We also added this text in the limitations; section 5.1, line 427ff:

*"It should be noted that quantified yields, irrigation amounts, and water deficits are subject to uncertainties due to model inputs, model parameters and model structure.*

*Some uncertainty arises from the use of a pedotransfer function (PTF) derived from point measurements for larger-scale applications. This scaling issue is discussed in the work of Weber et al. (2024) who also point out, that the transferability is highly dependent on the quality of the input data and the use of other environmental predictors. Due to the lack of sufficient observed soil moisture data, we were not able to validate the estimated soil hydraulic properties from the pedotransfer function. However, we still have confidence in the approach as the underlying soil map was derived by using a thoroughly trained and validated model and a large sample size (Swiss Competence Centre for Soil (KOBO) 2023)Also, the particular pedotransfer function we used in this study was already used and validated by Turek et al. (2023) on a typical Swiss agricultural soil.*

*Future work could investigate in detail how the different uncertainty sources impact the overall model output, e.g., by simulating parameter ensembles rather than single best parameter sets."*

3) Regarding the statement in the introduction (line 110):

*"To our knowledge, no study has simulated irrigation demand while considering restrictions on water supply and the impact of these on yield for temperate regions."*

We agree that we cannot make this statement with certainty. Therefore, we will remove it from the revised version of the manuscript.

We hope that the additional information provided here and in the manuscript, as well as a more detailed explanation of our reasoning and approach, will dispel any remaining doubts.